# Mono-Rhamnolipid Biosurfactants Synthesized by *Pseudomonas aeruginosa* Detrimentally Affect Colorectal Cancer Cells

**DOI:** 10.3390/pharmaceutics14122799

**Published:** 2022-12-14

**Authors:** Matthew S. Twigg, Simms A. Adu, Suguru Sugiyama, Roger Marchant, Ibrahim M. Banat

**Affiliations:** 1Pharmaceutical Science Research Group, Biomedical Science Research Institute, Ulster University, Coleraine BT52 1SA, UK; 2The Nutrition Innovation Centre for Food and Health (NICHE), School of Biomedical Sciences, Faculty of Life and Health Sciences, Ulster University, Coleraine BT52 1SA, UK

**Keywords:** biosurfactant, mono-rhamnolipid, anticancer, colorectal cancer, *Pseudomonas aeruginosa*

## Abstract

Over the past 15 years, glycolipid-type biosurfactant compounds have been postulated as novel, naturally synthesized anticancer agents. This study utilized a recombinant strain of *Pseudomonas aeruginosa* to biosynthesize a preparation of mono-rhamnolipids that were purified via both liquid and solid-phase extraction, characterized by HPLC-MS, and utilized to treat two colorectal cancer cell lines (HCT-116 and Caco2) and a healthy colonic epithelial cell line CCD-841-CoN. Additionally, the anticancer activity of these mono-rhamnolipids was compared to an alternative naturally derived anticancer agent, Piceatannol. XTT cell viability assays showed that treatment with mono-rhamnolipid significantly reduced the viability of both colorectal cancer cell lines whilst having little effect on the healthy colonic epithelial cell line. At the concentrations tested mono-rhamnolipids were also shown to be more cytotoxic to the colorectal cancer cells than Piceatannol. Staining of mono-rhamnolipid-treated cells with propidium iodine and acridine orange appeared to show that these compounds induced necrosis in both colorectal cancer cell lines. These data provide an early in vitro proof-of-principle for utilizing these compounds either as active pharmaceutical ingredient for the treatment of colorectal cancer or incorporations into nutraceutical formulations to potentially prevent gastrointestinal tract cancer.

## 1. Introduction

A malignant growth that develops either in the colon and/or rectum is defined as colorectal cancer [1]. The International Agency for Research of Cancer’ Global Cancer Observatory data show that the estimated age-standardized incidence rate for colorectal cancer in 2020 was 19.5. This places colorectal cancer as the fourth most prevalent form of cancer globally [2]. With estimated age-standardized mortality rates of 11.4 and 12.4, respectively, colorectal cancer is the second most common cause of cancer-related death in the United Kingdom and Republic of Ireland [2,3,4,5]. The primary strategy for the treatment of colorectal cancer is dependent on the stage of disease progression but is normally surgery, followed up by adjuvant chemotherapy [6]. Several medications are available for colorectal cancer chemotherapy; however, they are often nonselective against cancer cells alone and, as such, can lead to complications, which include cardio- and neurotoxicity, suppressed immune-activity, cancer recurrence and metastasis [7,8]. Consequently, finding novel active pharmacological ingredients for the treatment of colorectal cancer with higher therapeutic efficacy that will target cancerous and not healthy cells, leading to fewer adverse side effects is the “holy grail” of cancer research. As such an increasing body of research is focusing on the investigation of natural products that will target epithelial neoplastic cells in the gastrointestinal tract that will delay the progression of colorectal cancers [9,10,11,12]. One of such group of naturally derived compounds that are being actively investigated are microbial biosurfactants.

Microbial biosurfactants are amphiphilic secondary metabolites synthesised by bacterial and fungal species that possess surface-active properties [13]. Biosurfactants are primarily classified based on their molecular structure and species of origin [14,15]. Due to multiple physio-chemical properties of biosurfactants, they are utilised in a wide verity of industrial sectors, including medical applications [16,17,18,19,20]. Rhamnolipids (RLs) are a class of glycolipid type biosurfactants, which are among the most well-characterized to science [15]. The most well-known producer of RLs is the Gram-negative bacterium *Pseudomonas aeruginosa*, however, recent research has shown other bacterial species to be also capable of RL biosynthesis [21,22,23,24]. Individual RL congeners possess significant chemical variation with a common structure consisting of one or two hydrophilic rhamnose sugar moieties bonded to hydrophobic β-hydroxy fatty acids, which vary in length between 8 to 16 carbons and can be either saturated or unsaturated. RL congeners that consist of one rhamnose sugar are termed mono-rhamnolipids (mono-RLs) while congeners possessing two rhamnose moieties are termed di-rhamnolipids (Di-RLs) [25,26]. RLs are biosynthesised by three sequential steps with each step catalysed by a specific enzyme. The first step is the synthesis of fatty acid dimers from 3-hydroxy acyl-ACP, which is, in turn, derived from the fatty acid de novo synthesis pathway. This step is catalysed by rhamnosyltransferase-A (RhlA) encoded by the *rhlA* gene [21,27]. The second step, which yields mono-rhamnolipids, is the addition of dTDP-L-rhamnose to the fatty acid dimers, this reaction is catalysed by rhamnosyltransferase-B (RhlB) encoded by the *rhlB* gene [21]. Finally, the third step utilizes mono-rhamnolipids as a substrate adding a second dTDP-L-rhamnose to synthesize di-rhamnolipids. This reaction is catalysed by rhamnosyltransferase-2 (RhlC), encoded by the *rhlC* gene [28]. In *P. aeruginosa*, the first two genes (*rhlA* and *rhlB*) have been reported to be present on the bacterial chromosome in a single operon. The third gene, (*rhlC*), is located in a separate region of the bacterial chromosome from this operon [21,29]. The first two genes and *rhlC* are separately regulated at the transcriptional level by an acyl-homoserine lactone mediated quorum sensing system [21,30].

Several reports have been published in recent years on the anticancer properties of RLs, particularly on their selective cytotoxicity effects against cancer cells. Thanomsob and colleagues demonstrated that Rha-Rha-C_10_-C_10_ isolated from *P. aeruginosa* B189 mediated considerable toxicity on human breast cancer cell line MCF-7 [31]. Sanjivkumar and colleagues showed that partially purified RLs also possessed cytotoxic effects on colon (HT-29) and cervical (HeLa) cancer cell lines [32]. However, these studies either did not test the effects of RLs on cell lines isolated from normal tissue or used RL preparations that were impure or poorly characterized. In 2021, Semkova and colleagues observed that RL congeners also isolated from *P. aeruginosa* possessed cytotoxic effects against breast cancer cell lines MCF-7 and MDA-MB-231 while still exhibiting a cytotoxic effect on normal breast cell lines MCF-10a but at a statistically lower level [33]. The cytotoxic effects of RLs on four cancer cell lines (HepG2, Caco2, HeLa and MCF-7) and two normal cell lines (human HK-2 and primary rat hepatocytes) showed that both cancer and healthy cells were negatively affected by RLs due to their surface tension reducing effects [34]. Recently, a few studies have indicated that modification in the chemical structure of a biosurfactant can have significant effect with regards to anticancer properties. Sophorolipids (SLs), a type biosurfactant belonging to the same glycolipid class as RLs, were shown to be cytotoxic to colorectal cancer cell lines Caco2, HCT-116 and HT-15. However, a small deviation in molecular structure between either lactonic or acid congeners rendered a significant change in the effect on healthy gut epithelia, with lactonic-SLs being cytotoxic and acidic-SLs having no discernible effect [12,35]. Conversely, when applied to the melanoma cell line SK-MEL-28, lactonic-SLs were found to have a significantly higher cytotoxic effect than on the healthy keratinocyte cell line HaCaT [11]. Interestingly, the same study showed that at high concentration mono-RLs were significantly more cytotoxic to SK-MEL-28 than to the healthy HaCaT cell line, a result also echoed in breast cancer cell lines [11,33]. Based upon the two papers published by Callaghan in 2016 and 2022 showing colorectal cells are differentially affected by altering the biosurfactants molecular structure and those of Adu and colleagues in 2022 demonstrating cytotoxic effects of mono-RL congeners on melanoma cells compared to healthy keratocytes, we hypothesized that mono-RL would detrimentally affect colorectal cancer cells at a greater level than healthy gut epithelial cells [11,12,35].

To investigate this hypothesis, mono-RLs were selectively synthesised using a transposon mutant strain of *P. aeruginosa* deficient in the rhamnosyltransferase-2 expressing gene *rhlC*. In contrast to several previous studies using RLs to treat cancer cell lines, the chemical profile of the purified mono-RLs biosynthesized by the *P. aeruginosa rhlC* mutant were fully characterized via High-Performance Liquid Chromatography—Mass Spectroscopy (HPLC-MS). The physical activity of the extracted mono-RLs was determined via the measurement of their ability to reduce surface tensions. The cytotoxic effects of purified and fully characterized mono-RL congeners on two human colonic cancer cell lines (HCT-116 and Caco2) and a normal gut epithelial cell line CCD-841-CoN was investigated. In addition, these effects were compared to those elicited at the same concentration by Piceatannol, a potential new anti-colorectal cancer agent that is also derived from a natural source.

## 2. Materials and Methods

### 2.1. Bacterial Strain, Cell Lines and Culture Conditions

*P. aeruginosa* PAO1-∆*rhlC* was obtained from the University of Washington State, USA and routinely cultured in Oxoid Nutrient Broth (NB) (Thermo Fisher Scientific, Waltham, MA, USA) at 37 °C with shaking at 220 rpm.

Human colorectal cancer cell lines HCT-116 (ATCC^®^ CCL-247™, Manassas, VA, USA), Caco2 (ATCC^®^ HTB-37™), and human colonic epithelium cell line CCD-841-CoN (ATCC^®^ CRL-1790™) were kindly gifted by Dr. Breedge Callaghan, Ulster University, UK. HCT-116 and Caco2 were routinely cultured in Dulbecco’s Modified Eagle Medium (DMEM) 4.5 g L^−1^ D-glucose, L-glutamine (Thermo Fisher Scientific) supplemented with 10% (*v*/*v*) foetal bovine serum (FBS) (Thermo Fisher Scientific). CCD-841-CoN cells were routinely cultured in Minimal Essential Media (MEM) L-glutamine (Thermo Fisher Scientific) supplemented with 10% (*v*/*v*) FBS (Thermo Fisher Scientific). Incubation of all human cell lines was at 37 °C in a humidified environment containing 5% CO_2_. Cells were cultured until ≈80% confluency before splitting and/or experimental usage.

### 2.2. Polymerase Chain Reaction (PCR) and DNA Sequencing

The phylogenetic identity of *P. aeruginosa* PAO1-∆*rhlC* was confirmed by PCR amplification and subsequent sequencing of the gene encoding the 16S sub-unit of ribosomal RNA (16S rRNA). Confirmation of the presence of mono-RL biosynthesis genes *rhlA*/*rhlB* and absence of di-RL biosynthesis gene *rhlC* was carried out by PCR screening. Primer pairs for all PCR reactions are provided in Appendix A. DNA was extracted from stationary phase cultures of *P. aeruginosa* PAO1-∆*rhlC* using a DNeasy UltraClean Microbial Kit (50) (Qiagen, Hilden, Germany) used as per manufactured instructions. Exactions were quantified and assessed for purity by measuring absorbance at 260 and 280 nm using at NanoDrop ND-100 spectrophotometer (Thermo Fisher Scientific).

All PCR reactions were carried out in a TC5000 thermocycler (Techne, Ramsey, MI, USA) using Taq DNA polymerase (Thermo Fisher Scientific). Individual PCR reactions were made as per manufactures instructions to a volume of 50 μL with each containing 50 ng of template DNA. PCR reactions consisted of one cycle at 94 °C for 3 min (initial denaturation) followed by 30 cycles of 94 °C for 45 s (denaturation); specific primer annealing temperature (Appendix A) for 30 sec (annealing); and 72 °C for 90 s/kb of target DNA (extension), this was followed by one cycle at 72 °C for 10 min (final extension) and a holding temperature of 10 °C.

Amplicons were separated by molecular weight alongside a TrackIt 1 Kb DNA ladder (Thermo Fisher Scientific) on a 1% (*w*/*v*) agarose gel made with 1 X TBE buffer (Thermo Fisher Scientific). DNA was visualised under UV light using SybrSafe DNA stain (Thermo Fisher Scientific). Successfully amplified DNA was purified using a Wizard SV Gel and PCR Clean-Up System (Promega, Madison, WI, USA) used as per manufactures instruction and sequenced via the Sanger method by Eurofins Genomics (Cologne, Germany).

### 2.3. Mono-RL Biosynthesis, Extraction, and Purification

Primary seed cultures of *P. aeruginosa* PAO1-∆*rhlC* (5 mL) were grown under standard culture conditions to a stationary growth phase. These primary seed cultures were then utilized to inoculate 50 mL secondary seed cultures (1:10 *v*/*v*) in 500 mL Erlenmeyer flasks. Secondary seed cultures were grown under standard conditions to an exponential growth phase with bacterial growth monitored by measurement every 60 min of Optical Density (OD) at 600 nm using a Ultrospec 2000 spectrophotometer (Pharmacia Biotech, Piscataway, NJ, USA). At exponential growth phase, the secondary seed cultures were used to inoculate 500 mL of NB supplemented with 1% (*v*/*v*) glycerol (Merck, Rahway, NJ, USA) (1:10 *v*/*v*) in 2000 mL Erlenmeyer flasks. These cultures were grown at 37 °C for 96 h, on rotary shaking platform set at 220 rpm. Throughout growth the surface tension of the culture was monitored by the Du Noüy ring method using a K10ST digital tensiometer (Krüss, Hamburg, Germany) [13,36].

At the termination of the culture, RLs were extracted using liquid-phase extraction previously described by Twigg and colleagues with slight modifications [23]. Briefly, after centrifugation of culture broth at 1000× *g* for 30 min, the supernatant was separated and acidified to pH 2.0 with 1 M HCl (Merck). Mono-RLs were then extracted 3 × from the acidified supernatant with an e qual volume of HPLC-grade ethyl acetate (Merck). The organic phase was retained, pooled, and dried under vacuum at 40 °C with a rotary evaporator. The crude extract was resuspended in 10 mL HPLC-grade methanol (Merck), transferred to a scintillation vial and dried under N_2_ gas.

The resulting crude extract consisting of the mono-RLs, other extracted secondary metabolites and broth contaminants was purified via a solid phase extraction process using Strata SI-1 Silica (55 μm, 70 Å) Giga tubes 8B-S012 -KDG (Phenomenex, Torrance, CA, USA). Mono-RLs were eluted from the column with chloroform: methanol (5:0.5 *v*/*v*), dried under vacuum at 40 °C with a rotary evaporator, resuspended in 10 mL methanol, transferred to a scintillation vial and dried with N_2_ gas. Extraction yields were determined by gravimetric measurement [24].

### 2.4. Critical Micelle Determination

The Critical Micelle Concentration (CMC) of the purified mono-RLs was determined using the afore mentioned Du Noüy ring method [36]. In triplicate, a two-fold serial dilution series of the purified mono-RL was generated in distilled H_2_O ranging from 1000 μg mL^−1^ to 1.95 μg mL^−1^. The surface tension of each standard was determined using a K10ST Digital Tensiometer (Krüss). Using Prism 9 for Mac OS v9.4.1 (458) (GraphPad Software, San Diego, CA, USA), CMC value was determined by plotting surface tension measurement against mono-RL concentration, (plotted on a Log10 scale), followed by Sigmoidal dose-response (variable slope) analysis [13].

### 2.5. HPLC-MS Analysis of RLs

Purified mono-RL preparations utilized in this study were fully characterized for congener composition using HPLC-MS. MS analysis was carried out using a LCQ Classic electrospray ion-trap spectrometer (Thermo Fisher Scientific) as described in detail by previous research [11,37].

### 2.6. Cytotoxicity Assessment

In vitro cytotoxic activity of Mono-RL and Piceatannol (kindly donated by the School of Pharmacy, Ulster University) was assessed using and a Cell Proliferation Kit II (XTT) (Roche, Basel, Switzerland). HCT-116, Caco2 and CCD-841-CoN were grown to confluency in complete medium, seeded in 96-well plates (Sarstedt, Hildesheim, Germany) at a density of 1 × 10^4^ cells per well in 100 μL complete medium and incubated for 24 h. Following initial incubation, the cells were serum-starved for a further 24 h. The cells were then treated for 24 h with 100 μL per well serum-free medium supplemented with either mono-RL or Piceatannol ranging from 10–100 μg mL^−1^; serum-free medium supplemented with 1% (*v*/*v*) HPLC-grade methanol (vehicle control), serum-free medium supplemented with 2% triton (positive control) or un-supplemented complete media. Following treatment culture medium was aspirated, the cells washed with sterile Phosphate Buffered Saline (PBS) (Thermo Fisher Scientific). XTT labelling mixture was prepared as per manufactures instructions and added to cells at a volume of 50 μL per well. XTT-loaded plates were incubated at 37 °C for 3 h. Following incubation absorbance at 450 nm and 650 nm was measured with a FLUOstar Omega plate reader (BMG LABTECH, Ortenberg, Germany). All experiments were carried out in triplicate with six technical replicates per treatment condition.

### 2.7. Morphological Examination

The morphological effects of treatment with either mono-RL or Piceatannol was assessed by morphological examination of cells post-treatment using visible light bright field microscopy. HCT-116, Caco2 and CCD-841-CoN were grown to confluency in complete medium, seeded into 24-well plates (Sarstedt) at 5 × 10^4^ cells per well in 500 μL complete medium and incubated for 24 h at 37 °C. Following initial incubation, the cells were serum-starved for a further 24 h. The cells were then treated for 24 h with 500 μL per well serum-free medium supplemented with either mono-RL or Piceatannol at 60 μg mL^−1^; serum-free medium supplemented with 1.5% (*v*/*v*) HPLC-grade methanol (vehicle control) or un-supplemented complete media (negative control). Following treatment, cells were directly imaged at 200× magnification using a Digital Sight DS-5M (Nikon, Tokyo, Japan) attached to an Eclipse TS100 microscope (Nikon). All experiments were carried out in triplicate with three technical replicates per treatment condition. Each replicate was imaged three times and images for publication were randomly selected for publication using a computer. Scale bars (100 μm) were added to images using ImageJ 1.53t software [38].

### 2.8. Acridine Orange and Propidium Iodine Staining

To assess the morphological pattern of cell death induced by mono-RL and Piceatannol treated cells an acridine orange (AO)/propidium iodine (PI) staining technique was utilized [39]. HCT 116, Caco2 and CCD-841-CoN cells were cultured and treated as previously described in Section 2.7. Following treatment, cells were washed three times with sterile PBS (Thermo Fisher Scientific) and subsequently incubated for 3 min with 100 μg mL^−1^ AO and PI (Merck Life Science, Bengaluru, India) mixed at a ratio of 1:1. Cells were then re-washed with sterile PBS (Thermo Fisher Scientific) and stained cells were imaged at 200× magnification using an Eclipse TS100 fluorescence microscope (Nikon). The excitation and emission wavelengths for AO were 493 nm and 535 nm, and for PI were 535 nm and 614 nm. All experiments were carried out in triplicate with three technical replicates per treatment condition. Each replicate was imaged three times and images for publication were randomly selected for publication using a computer. Scale bars (100 μm) were added to images using ImageJ 1.53t software [38].

### 2.9. Statistical Analysis

Statistical analysis of cell viability data was achieved using a two-way Analysis of Variance (ANOVA), followed by Tukey’s post hoc testing. A value of *p* ≤ 0.05 was considered statistically significant. All statistical analysis was carried out using Prism 9 for Mac OS v9.4.1 (458) (GraphPad Software, San Diego, CA, USA).

## 3. Results

### 3.1. PCR Confirmation of P. aeruginosa PAO1 ∆rhlC Strain Identity and Mutation

DNA amplified from *P. aeruginosa* PAO1 ∆*rhlC* using universal primers for the 16S rRNA reference gene was sequenced and compared via pair-wise sequence analysis to the 16S rRNA sequence of *P. aeruginosa* PAO1 published in the NCBI Nucleotide database. The pair-wise sequence alignment showed a 100% sequence match to the published sequence of *P. aeruginosa* PAO1. Confirmation of *rhlC* mutation was carried out via PCR screening using primers internal to each RL biosynthesis gene. Following PCR using genomic DNA extracted from *P. aeruginosa* PAO1 ∆*rhlC* as a template, DNA fragments whose molecular size corresponded to the expected amplicon size of both *rhlA* and *rhlB* (998 bp and 914 bp, respectively) was observed (Appendix A). PCR screening using primers designed to amplify *rhlC* failed to generate a DNA fragment of the expected amplicon size of 655 bp (Appendix A).

### 3.2. Molecular Profile and Physical Analysis of RLs Synthesised by P. aeruginosa PAO1 ∆rhlC

To generate mono-RLs for analysis *P. aeruginosa* PAO1 ∆*rhlC* was cultured for 96 h in NB supplemented with glycerol. Exponential bacterial growth was observed during the first 24 h of culture, which corresponded to a significant reduction in culture supernatant surface tension from 60 mN m^−1^ to 28 mN m^−1^. Surface tension then remained relatively constant throughout the remaining portion of the growth cycle (Appendix A). As expected, HPLC-MS analysis of RL congeners extracted and purified from cell-free culture supernatant at the conclusion of the growth cycle showed only mono-RL to be present (Figure 1). In the purified extract the relative abundance of mono-RL was 96.6% with the remaining 3.4% of the molecules detected being attributed to contaminants that could not be removed by solid-phase extraction. The relative quantification of each mono-RL congener synthesized by *P. aeruginosa* PAO1 ∆*rhlC* showed Rha-C_10_-C_10_ to be the predominant congener present (74.59%), followed by Rha-C_10_-C_12_ (7.62%), (Table 1). The yield of purified mono-RL obtained during the growth of *P. aeruginosa* PAO1 ∆*rhlC* in this study was 2.350 g L^−1^ of cell-free supernatant. The surface tension reducing properties of the purified mono-RLs were assessed and critical micelle concentration was determined to be 45.6 μg mL^−1^.

### 3.3. Effects of Mono-RL on the Viability of Colorectal Cancer Cell Lines & Healthy Gut Epithelia

The cytotoxicity of both mono-RLs and Piceatannol on colorectal cancer cell lines HCT-116 and Caco2 was assessed using a XTT cell viability assay. Following 24 h of mono-RL treatment viability of both HCT-116 and Caco2 cell lines was reduced to a range of 14.99 to 12.16 percent of untreated cells at concentrations between 10 and 100 μg mL^−1^. This was a statistically significant reduction when compared to the vehicle-only (1.5% *v*/*v* methanol) treated cells (*p* ≤ 0.0001, Figure 2A). Viability of the mono-RL treated healthy colonic epithelial cell line CCD-841-CoN was only significantly reduced in comparison to vehicle-only treated cells at both 90 or 100 μg mL^−1^, with viability only reduced to 79.92 and 80.50 percent of the untreated cells (*p* ≤ 0.0001). In comparison to untreated cells, treatment with the delivery vehicle (1.5% *v*/*v* methanol) did not significantly affect cell viability in any of the three cell lines tested (*p* ≥ 0.05, Figure 2A). A comparison of cell viability in the two colorectal cancer cell lines treated with mono-RL to the healthy colonic epithelial cell line treated under the same conditions showed that there was a significant reduction in both the HCT-116 and Caco2 cell lines at all concentrations tested (*p* ≤ 0.0001, Figure 2A).

Treatment of the Caco2 and CCD-841-CoN cell lines with Piceatannol resulted in no significant reduction in cell viability when compared to the vehicle-only treated cells (*p* ≥ 0.05, Figure 2B). Except for both the 90 and 100 μg mL^−1^ treatment concentrations, HCT-116 cells lines treated with Piceatannol also resulted in no significant reduction in viability when compared to the vehicle-only treated cells (*p* ≥ 0.05). Treatment with both 90 and 100 μg mL^−1^ Piceatannol resulted in a significantly increased cell viability when compared to the vehicle-only treated cells (*p* = 0.0390 and 0.005 respectively, Figure 2B). At all concentrations viability of both HCT-116 and Caco2 was significantly reduced when treated with mono-RL in comparison to Piceatannol (*p* ≤ 0.0001). The positive control for the assay (100 μg mL^−1^ Triton X) resulted in a significant reduction in cell viability of all three cell lines (*p* ≤ 0.0001, Figure 2A,B).

### 3.4. Morphological Assessment of Colorectal Cancer Cell Lines & Healthy Gut Epithelial Cells Treated with Mono-RL and Determination of Cell Death Mechanisms

Direct imaging, using bright field microscopy, was used to assess the effect treatment with both mono-RL and Piceatannol, (at a concentration midrange to that used in cell viability assays), had on the morphology of both the colorectal cancer cell lines (Caco2 and HCT-116) and the normal colon epithelial cell line (CCD-841-CoN). No visible difference was observed in each of the three cell lines treated with 60 μg mL^−1^ Piceatannol compared to those either untreated or treated using the vehicle control, with the cells maintaining a monolayer of expected morphology (Figure 3). However, in both the colorectal cancer cell lines treated with 60 μg mL^−1^ mono-RL, the cells were observed to have a rounded morphology with many cells detaching from the surface culture flask into the medium (Figure 3). This effect was not observed in the normal colon epithelial cell line treated with 60 μg mL^−1^ mono-RL, with the cells maintaining a normal morphology and growing in a monolayer similar to that of the untreated and vehicle-only control treated cells (Figure 3).

The potential mechanism of cell death in each of the three cell lines treated under all experimental conditions was assessed using staining with both AO and PI. No staining with PI was observed in untreated and vehicle control-treated cells. These untreated and vehicle control treated cells had no reduction in cell population; cells were mainly shown to omit green fluorescence after staining, indicating the cells were morphologically viable (Appendix A).

Treatment with mono-RL resulted in drastic reduction in cell population of Caco2 and HCT-116 cell lines. The few cells adhering to bottom of plates had their membranes disrupted while maintaining red/orange granular nuclei indicating necrotic cell death (Appendix A). Conversely, CCD-841-CoN cells treated with mono-RLs appeared to be morphologically viable (Appendix A). Similarly, treatment with Piceatannol had no effects on cell population, morphology, and membrane disruption of cell lines utilized in this study (Appendix A).

Using the images of the stained cells, the percentage distribution of necrotic to live cells was calculated. In all three cells lines there was no significant difference in the percentage distribution of necrotic cells between untreated cells compared to those treated with either the vehicle control (1.5% *v*/*v* methanol) or 60 μg mL^−1^ Piceatannol (*p* ≥ 0.05, Figure 4). There was however a significant increase in the percentage distribution of necrotic colorectal cancer cells (HCT-116 and Caco2) treated with 60 μg mL^−1^ mono-RL compared to untreated colorectal cancer cells (*p* ≤ 0.0001, Figure 4). Likewise, there was a significant increase in the percentage distribution of necrotic cells in both colorectal cancer cell lines treated with 60 μg mL^−1^ mono-RL compared to the mono-RL treated healthy colonic epithelial cell line (CCD-841-CoN) (*p* ≤ 0.0001, Figure 4). However, no significant difference in percentage distribution of necrotic cells was observed between mono-RL and untreated healthy colonic epithelial cells (CCD-841-CoN) (*p* = 0.9561, Figure 4).

## 4. Discussion

The study presented here attempted to assess potential anticancer properties of naturally synthesized microbial mono-RL congeners. The anticancer effects of both mono-RLs and other forms of microbial biosurfactants have been investigated for the past 15 years in a multitude of cancer cell lines isolated across the body [40]. The selection of colorectal cancer for this current study was based on two salient points; the first is that colorectal cancer is a significant cause of global cancer-related mortality [2], the second is that the colon, and as an extension the upper gastro-intestinal tract, is easily accessible to formulations comprised of microbial biosurfactants delivered orally or via the anus. As such, these compounds could be included in medications as active pharmaceutical ingredients for the direct treatment of colonic cancer or potentially added to nutraceutical products for the prevention of gastrointestinal tract cancer. The current treatment strategy for colorectal cancer is surgery followed up by chemotherapy [41]. Most chemotherapy treatments are detrimental to both cancer and healthy tissues, as such identifying compounds possessing activity specifically targeted at tumour cells is a key requirement for new anticancer treatments. Cell viability assays carried out here demonstrated that mono-RLs biosynthesized by *P. aeruginosa* PAO1 ∆*rhlC* were significantly cytotoxic to two cell lines originally isolated from colorectal tumours (HCT-116 and Caco2). The cytotoxicity of rhamnolipid-type biosurfactants has been previously reported [31,34,42]. However, many of these studies utilized biosurfactant preparations that were ether not sufficiently pure or were characterized using techniques other than HPLC-MS, the gold standard techniques for the analysis of glycolipid biosurfactants [13,43,44,45]. Additionally, many of these studies either lacked a healthy cell line comparison or utilized a comparative healthy cell line from a separate area of the body [31,44,46,47].

Cell viability data presented here showed that the treatment of a healthy colonic epithelial cell line (CCD-841-CoN) with the same mono-RLs caused no significant detrimental effects. Direct imaging of the treated cell lines reflected cell viability assays, with mono-RL treated colorectal cancer cells shown to be either detached from the surface of the culture plates or possessing an un-healthy morphology. In contrast the mono-RL treated healthy colonic epithelial cell line was discernibly unaffected. Together these data suggests that the cytotoxic effects of the mono-RL preparation were limited to the cancer cell lines. Investigation of the mechanisms of cell death induced in colorectal cancer cells by treatment with mono-RL was investigated by AO/PI staining. The results from the staining of mono-RL treated cells showed that cell death was the result of necrosis and not apoptosis. This appears to be consistent with the findings of Callaghan and colleagues in the treatment of the same cell lines with both acidic and lactonic sophorolipids [12,35]. We postulate that the surface-active properties of the mono-RLs are having a disruptive effect on the cancer cell membranes inducing necrotic pathways of cell death. Notwithstanding, it is a generally accepted that for effective anticancer agent, induction of gentler mechanism of cell death such as apoptosis will be favourable over necrosis in the clearance of colorectal neoplastic cells, particularly in in vivo systems. Hence, further in vitro and in vivo studies aimed at investigating/ascertaining a gentler mechanism of cell death will be necessary.

Interestingly the significant reduction in cell viability of colorectal cancer cell lines treated with mono-RL contrasts with previous work using these congeners to treat both breast cancer cell lines and melanoma cells. In 2013 Zhao and colleagues reported that it was di-RL and not mono-RL responsible for cytotoxicity when treating the breast cancer cell lines MRC-7 [42]. More recently, Adu and colleagues showed that treatment of the SK-Mel-28 melanoma cell line with mono-RLs at the same concentration range as was used in this study (10–100 μg mL^−1^) had no effect on viability, and it was not until mono-RL concentration was raised to above 400 μg mL^−1^ that one observed a significant fall in viability [11]. The differential effects glycolipids have on cancer cell lines derived from separate areas of the body has also been observed when treating both melanoma and colorectal cancer cells with sophorolipids with acidic sophorolipids causing a targeted cytotoxic effect on colorectal cancer cells whilst having no effect on melanoma cells [11,12].

These contrasting results appears to suggest the anticancer effect of glycolipid congeners is not universal for all types of cancer and that the specific cancer cell line or area of the body where the cell line was isolated from is of importance. As Adu and colleagues demonstrated that di-RLs had a cytotoxic effect on melanoma cells at concentrations above 40 μg mL^−1^ and Zhao and colleagues showed cytotoxicity of di-RLs in MRC-7 cells, it would be interesting to progress this work by treating colorectal cancer cells with these congeners, however 100% pure preparations or di-RLs are more difficult to produce [11,42]. The differential effects mono-RL congeners have on cell lines derived from separate types of cancer may be due to biophysical properties of the different cell membranes and how they interact with the physio-chemical properties of each different type of glycolipid congener type [48]. This may also explain why glycolipid congeners with minor modifications in molecular structure that effect their charge and hydrophobicity have differential effects on cells, as was observed in healthy colonic epithelial cells, which were detrimentally affected by treatment with lactonic sophorolipids but not acidic sophorolipids [12,35]. It would be of interest to assess if this is true for rhamnolipid type glycolipid congeners by treating both health colonic epithelial cells and colorectal tumour cells with preparations of di-RL.

Piceatannol is a new potential anticancer drug isolated from the roots of Japanese knotweed (*Polygonum cuspidatum*). Its structure is similar to Resveratrol, one of the most studied natural polyphenolic compounds [49]. Piceatannol has been reported to cause S phase arrest of HCT-116 cells and induce apoptosis by G0/G1 arrest in the bladder cancer cell line HT-1376 [50]. Based upon this evidence we compared the activity of Piceatannol to mono-RL congeners at the same concentrations, under the same experimental conditions. At the concentrations tested in this study, no significant effect on the viability either of the colorectal cancer cell lines was observed and imaging of stained Piceatannol treated cells revealed no evidence for the induction of apoptosis. This could be explained by the relatively short treatment time (24 h) as previous studies have suggested that cancer cell death was achieved following 72 h of co-incubation with 100 μM of Piceatannol [51]. Despite this, the data presented by this study appears to show that mono-RL congeners are far more efficacious than Piceatannol causing targeted cell death at both a lower concentration and in a shorter treatment period.

This study utilised the *P. aeruginosa* PAO1 ∆*rhlC* mutant strain to guarantee that rhamnolipids synthesized for testing would only possess a single rhamnose moiety [28]. HPLC-MS profiling of congeners purified from culture cell-free supernatants of this strain confirmed this, and that the purification protocol utilized here was stringent enough to generate a preparation of mono-RLs at a purity sufficient for in vitro testing (96.6%). The purity of biosurfactant preparations being utilised to determine any biomedical effect is an important factor in study design [13]. The level of purity deemed optimal for this study was determined based on previously published research investigating the effect of various glycolipid preparations had on both gastrointestinal cancer cells and melanoma cells [11,12,35]. The final yield of mono-RL produced by the ∆*rhlC* mutant for this study (0.6 g L^−1^) was low compared to that reported by several *P. aeruginosa* strains [52]. It should, however, be noted that RL yield significantly varies depending on the bacterial strain and the type and concentration of the carbon source used during bacterial fermentation [53]. In this study, little in the way of medium optimization was carried out and as such may have led to the low yield. Due to the low concentrations of mono-RLs required for experimental testing this low production yield was not a significant problem, and highlights that the usage of biosurfactants in biomedical applications such as anticancer treatments would not require extensive production capacity, a limiting factor in many other potential applications for microbial biosurfactant compounds [16]. In addition to media optimisation consideration may need to be given to replacing *P. aeruginosa* as the producer organism due to its potential pathogenic nature. Non-pathogenic, rhamnolipid producing wild-type Pseudomonad strains have been reported as have recombinant non-pathogenic strains of *E. coli* and *Pseudomonas putida* that express the rhamnolipid biosynthesis genes [23,54,55].

The congener profile present in the mono-RL preparation as revealed by HPLC-MS analysis were as expected based on previous reports of rhamnolipid biosynthesis by *P. aeruginosa* with prevalence of congeners with fatty acid chains possessing 10–12 carbons [26]. Additionally, the preparation was shown to have significant surface-active properties, reducing the surface tension of culture medium to approximately 27 mN m^−1^, again this was consistent with previously published research [23]. This poses yet another interesting question with regards to the biological activity of these compounds. Is anticancer activity significantly affected by the compound’s physio-chemical properties? This paper has already discussed the differential effect altering the hydrophilic moity of the glycolipid congener has on anticancer activity, however, little is known about the effects of altering the hydrophobic moiety of the congeners. Future studies may wish to assess the effect of treating both healthy colonic epithelia and colorectal cancer cells with RL congeners possessing acyl-sidechains that differ in length/saturation from those predominantly synthesized by *P. aeruginosa*. This could be achieved by using an alternative bacterium in the biosynthesis process such as *Burkholderia thailandensis*, a species known to synthesize RLs with acyl-sidechains of predominantly 14 carbons in length [22].

## 5. Conclusions

In conclusion, this study has suggested that purified mono-RLs biosynthesized by *P. aeruginosa* have potential anticancer properties within the gastrointestinal tract, and in in vitro testing, appears to have a higher efficacy than another potential anticancer drug Piceatannol. To be used as an anticancer agent, targeted specificity to cancer cells and not normal cells is a key factor and in the case of mono-RLs the results presented here appear to demonstrate this. The exact reason for this targeted activity is only postulated here and as such would be the subject of further research, especially considering comparative research that shows it is di-RLs that have the better anticancer activity when it comes to melanoma cells. The results presented here are an optimal proof-of-principle study for the usage of rhamnolipid-type microbial biosurfactants as anticancer agents in the gastrointestinal tract. To progress this work forward towards proof-of-concept and application, we plan to further investigate the mechanism of action by which these compounds elicit targeted cell death in the tumour cell lines. To achieve this, we plan to measure key cellular proteins involved in both necrosis and apoptosis and utilize metabolomic/lipidomic techniques. Following on from these in vitro investigations, we also plan to assess proof-of-concept for using these compounds as cancer treatment/preventative agents by assessing bioavailability and toxicity using appropriate ex vivo or in vivo models.

## Figures and Tables

**Figure 1 pharmaceutics-14-02799-f001:**
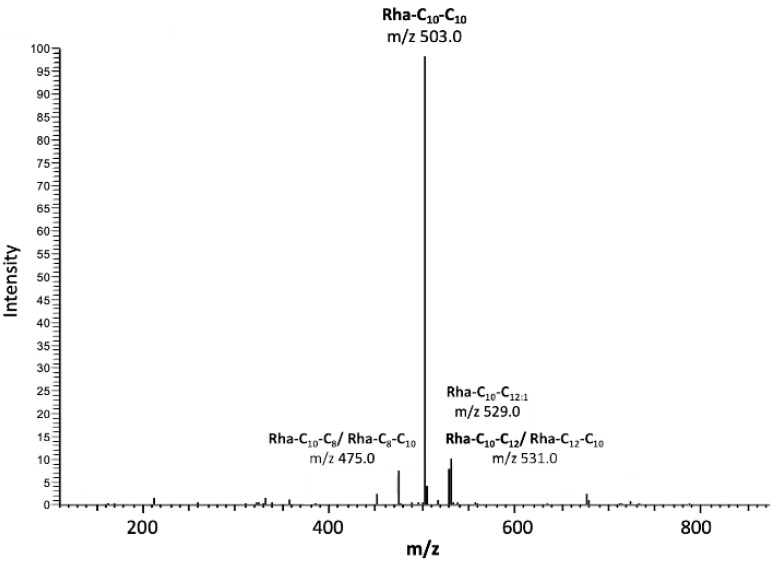
HPLC-MS profile of a SPE purified extract obtained from *P. aeruginosa* PAO1 ∆*rhlC* cell-free supernatants. Only peaks corresponding to mono-RL congeners were identified with the predominant peaks identified as Rha-C_10_-C_10_ and Rha-C_10_-C_12_.

**Figure 2 pharmaceutics-14-02799-f002:**
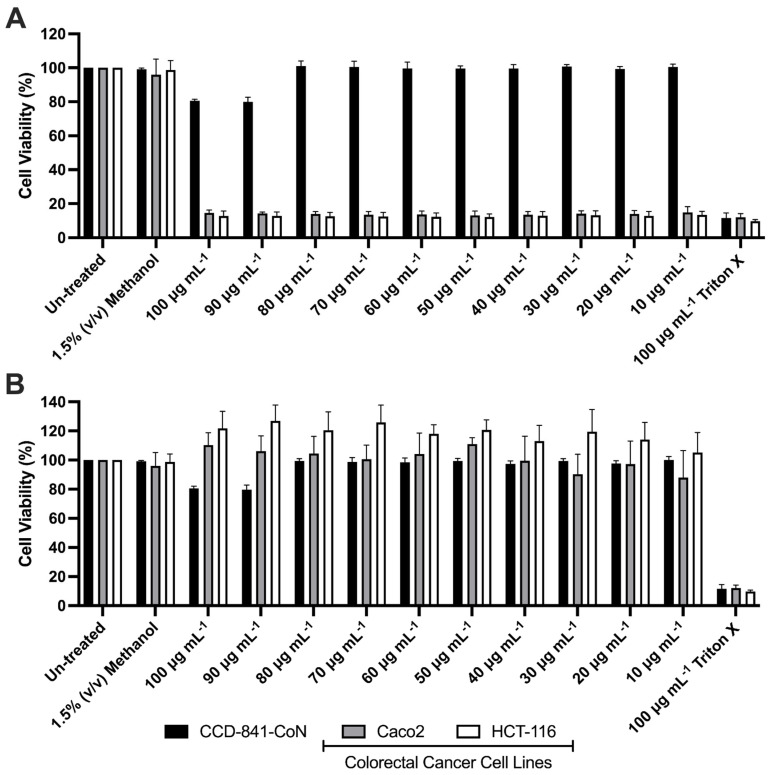
Mean cell viability of colorectal cancer cell lines Caco2/HCT-116, and healthy colonic epithelial cell lines CCD-841-CoN treated with 10–100 μg mL^−1^ mono-RL (**A**) and 10–100 μg mL^−1^ Piceatannol (**B**). Cells were also treated with 1.5% (*v*/*v*) methanol as a vehicle-only control and 100 μg mL^−1^ Triton X as a positive control. Cell viability assays were carried out three independent times, error bars represent standard deviation from the mean.

**Figure 3 pharmaceutics-14-02799-f003:**
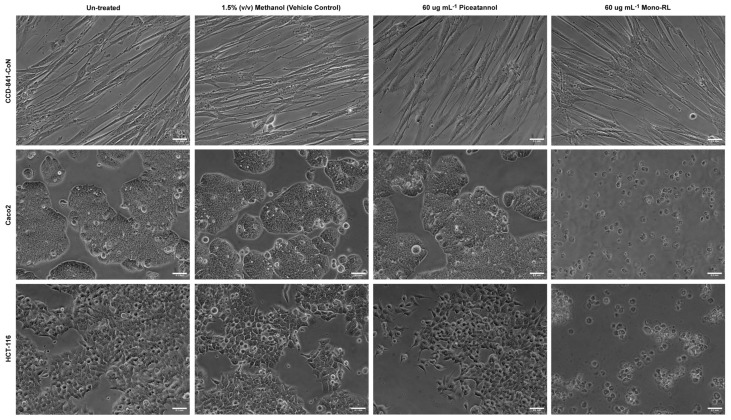
Randomly selected representative images of healthy colonic epithelial cell line CCD-841-CoN and colorectal cancer cell lines Caco2 and HCT-116 either untreated or following 24 h of treatment with 1.5% (*v*/*v*) methanol (vehicle control); 60 μg mL^−1^ Piceatannol; or 60 μg mL^−1^ mono-RL. Cells were imaged at 200× magnification, scale bar = 100 μm.

**Figure 4 pharmaceutics-14-02799-f004:**
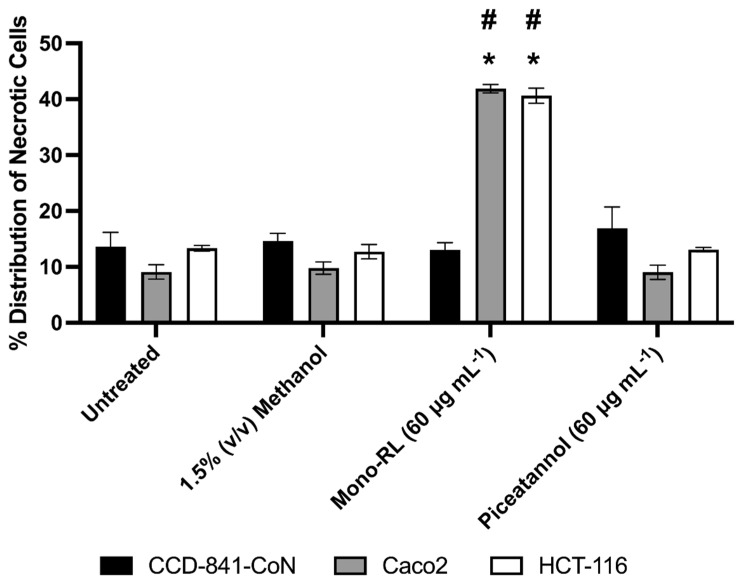
Mean percentage distribution of necrotic cells for each cell line following 24 h of each treatment condition was calculated from three randomly selected images; error bars represent standard deviation from the mean. * = significantly increased mean percentage distribution of necrotic cells compared to the untreated controls, # = significantly increased mean percentage distribution of necrotic cells in each cell line treated under the same conditions. (Two-ANOVA followed by Tukey’s multiple comparison test, *p* ≤ 0.05).

**Table 1 pharmaceutics-14-02799-t001:** Congener profile of mono-RLs extracted and purified from cell-free culture supernatant of *P. aeruginosa* PAO1 ∆*rhlC* as determined by HPLC-MS. The percentage relative abundance of each mono-RL congener is also provided.

Retention Time (Min)	*m*/*z*	M_w_	Relative Abundance (%)	Congener
20.93	333.0		1.07	Rha-C_10_
24.37	356.8		0.76	Rha-C_12:2_
17.62	475.0		5.60	Rha-C_10_-C_8_
20.78	503.0		74.59	Rha-C_10_-C_10_
22.47	529.0		5.89	Rha-C_10_-C_12:1_
23.87	531.0		7.62	Rha-C_10_-C_12_
24.37	557.0		0.27	Rha-C_10_-C_14:1_
26.09	559.1		0.22	Rha-C_12_-C_12_
21.97	517.0		0.59	Rha-C_10_-C_10_-CH_3_

## Data Availability

All data presented in this article will be made freely available upon a reasonable request to the corresponding author (Ibrahim Banat).

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
