# Peer review of "Mono-Rhamnolipid Biosurfactants Synthesized by Pseudomonas aeruginosa Detrimentally Affect Colorectal Cancer Cells"

_pharmaceutics, 2022, doi:10.3390/pharmaceutics14122799_

Round 1

Reviewer 1 Report

This work biosynthesized a preparation of mono-rhamnolipids by a recombinant strain of Pseudomonas aeruginosa and further verified the effectiveness of the mono-rhamnolipid in detrimentally affecting colorectal cancer cells (HCT-116 and Caco2). Based on the cytotoxicity assessment, the mono-rhamnolipid showed detrimental effect on colorectal cancer cells but not on healthy cell line. These results suggest that this ingredient may provide new formulation options for drugs related to gastrointestinal tract cancer. However, I request the following clarifications or modification.

1.      The determination of the potential anticancer activity mechanism of this ingredient only through staining experiments is too partial. Why are there no further studies at the protein expression level?

2.      It is well known that the pH level of the gastrointestinal environment and the composition of intestinal flora, for example, are complex. How to ensure that the potential bioactivity of this ingredient is still effective under such circumstances?

3.      The ingredient is derived from bacteria, and the purity used in the experiment is not 100%. Is there sufficient data to prove the biological safety of the ingredient and the strain (Pseudomonas aeruginosa) and its secretions in the experiment?

Author Response

Thank you for your comments. Kindly see below comments on the points mentioned.

  1. The determination of the potential anticancer activity mechanism of this ingredient only through staining experiments is too partial. Why are there no further studies at the protein expression level?

We thank the reviewer for their constructive criticism of our manuscript. As we state at the end of the manuscript, the study we present here was a preliminary proof-of-principle investigation of potential anti-cancer activity of mono-RL congeners. The study is complementary to several publications we have generated over the past 4 years investigating potential roles of glycolipid-type biosurfactants in cancer treatment. As such the aim of the study was to demonstrate that these congeners negativity effect a cancer cell lines that would be accessible for a biosurfactant delivered either topically, orally, or anally. In this regard we feel the results we present met the study’s aim. We also present cellular staining data postulating a potential mechanism of cell death that is induced by these congeners. We accept that this may be limiting, and we have now secured funding to further investigate the mechanisms of action using additional techniques such as the investigation of various protein levels using appropriate immunoassays and other techniques and hope to report these findings soon. Using the comments from the reviewers, we have made alterations in the conclusion section of the manuscript to provide a more detailed description of out plan to progress this work up the technology readiness scale, please see lines 533-543.

  1. It is well known that the pH level of the gastrointestinal environment and the composition of intestinal flora, for example, are complex. How to ensure that the potential bioactivity of this ingredient is still effective under such circumstances?

We do know that rhamnolipids are relatively stable in at various pH’s and that their effect on the intestinal flora is an interesting area of research being progressed. One of the best ways to assess this would be to utilise an appropriate in vivo animal model. We have done this before with similar glycolipid compounds and showed that the compounds were well tolerated and that they accessed the gut on a colorectal cancer mouse model (Callaghan et al. 2022). However, as this study was proof-of-principle in vivo testing was not justifiable either financially or ethically.

Callahan et al. 2022 - https://link.springer.com/article/10.1007/s00253-022-12115-6

  1. The ingredient is derived from bacteria, and the purity used in the experiment is not 100%. Is there sufficient data to prove the biological safety of the ingredient and the strain (Pseudomonas aeruginosa) and its secretions in the experiment?

As we have stated in our concluding remarks of the manuscript this study was a proof-of-principle study designed to assess the ability of these biologically produced compounds to selectively kill colorectal cancer cells. We have demonstrated this main aim in the data presented. The biological safety of the compounds would be a point to establish in the progression of this work. However, to try to address the reviewers comments we can state that the level of purity achieved in this study of 96.6% is very high. As we state in lines 288 and 289 the remaining 3.4% is likely to be media components such as fatty acids which are required to elicit biosurfactant production by the bacterium. This level of purity is far higher than that which is considered safe when these compounds are incorporated into naturally derived personal-care and laundry products that are currently available on the market. We also have evidence that similar compounds to the those tested here have been shown as safe for incorporation as emulsification agents in some food products in the United States, again at purity levels below what was achieved here. To increase purity further purification of individual mono-RL congeners using techniques such as HPLC could be a step forward and worth consideration in the progression of this technology. The significantly low concentration where we seen a targeted killing effect on the tumour cells makes this option increasingly viable. Finally in addressing the reviewers concern of the use of the bacterium Pseudomonas aeruginosa to synthesize the compounds, this is something we accept may have to change to allow progression of the technology due to the pathogenic nature of the species. This could be achieved by genetically modifying a non-pathogenic host strain such as Pseudomonas putida to produce mono-RLs. There is evidence in the that this has been successfully carried out resulting a good yield of compound.

Finally, an overall editing to improve language presentation and references included.

Reviewer 2 Report

The article “Mono-rhamnolipid biosurfactants synthesized by Pseudomonas aeruginosa detrimentally affect colorectal cancer cells” by M.S. Twigg and colleagues biosynthesized and characterized mono-rhamnolipids and investigated their properties in vitro. The article is well-written and organized, the weakest point, in my opinion, is the cellular characterization, which could have been more detailed thus providing interesting information about the biological effects of mono-rhamnolipids supplementation.

In particular, HPLC or NMR metabolomics/lipidomics would allow to better understand eventual differences in the mechanism of action between mono-RLs and piceatannol.

Also, NMR analysis of the RLs would provide a definitive confirmation of the chemical structures of the glycolipids.

These 2 additions would improve significantly the impact of the article.

Also, why did the authors used a relatively new drug like piceatannol and not a more common and well-characterized drug, such as cisplatin or doxorubicin, as a control drug for the experiments?

Finally, minor editing errors are present and must be corrected (e.g., missing dot at line 298).

The bibliography must be formatted properly according to the journal style (e.g. lines 84, 102, 103, 175, 184, etc.).

Rpm must be converted to either rcf or g (lines 124 and 171).

Line 192: please insert the reference to the original article.

Author Response

Please see below your comments and responses to these comments:

  • The article “Mono-rhamnolipid biosurfactants synthesized by Pseudomonas aeruginosa detrimentally affect colorectal cancer cells” by M.S. Twigg and colleagues biosynthesized and characterized mono-rhamnolipids and investigated their properties in vitro. The article is well-written and organized, the weakest point, in my opinion, is the cellular characterization, which could have been more detailed thus providing interesting information about the biological effects of mono-rhamnolipids supplementation. In particular, HPLC or NMR metabolomics/lipidomics would allow to better understand eventual differences in the mechanism of action between mono-RLs and piceatannol.

We thank the reviewer for their constructive criticism of our manuscript.

In responding to this comment, we would respectfully refer the editor and reviewer to our response to the first point made by reviewer 1. Having gained additional funding, it is our intension further investigate using methodologies such as NMR metabolomics/lipidomics. Using the comments from the two reviewers, we have made alterations in the conclusion section of the manuscript to provide a more detailed description of our plan to progress this work up the technology readiness scale, please see lines 533-543.

  • Also, NMR analysis of the RLs would provide a definitive confirmation of the chemical structures of the glycolipids.

The study was primarily concerned in having a preparation of glycolipids that was only comprised on mono-RL congers, this requirement was achieved by utilising a recombinant bacterial strain only capable of mono-RL biosynthesis the fermentative production process. Next the mono-RL generated by this strain were purified from cell-free supernatant using both liquid and solid phase extraction/ purification processes that we have developed over the course of 20 years working in the forefront of microbial biosurfactant technology (Primary reference for this Smyth et al. 2010). Finally, these purified congeners were analysed using HPLC-MS techniques again that we have specifically developed for RL characterisation (Rudden et al 2015, Smyth et al 2010). Using these methodologies allowed us to present (in this manuscript) a full breakdown of the different congeners present in our preparation and provide the percentage relative amounts of each congener in the preparation. This purification/ analytical pipeline was determined by leading researchers in the field of microbial biosurfactant technology as a “gold standard” approach to RL characterisation (Twigg et al, 2021, Zompra et al. 2022) and was similarly utilised in other recent publications relating to anti-cancer activity in glycolipids (Adu et al. 2022, Callahan et al. 2022). We did not therefore feel the need for NMR to provide further details of structure.

Please also see:

Smyth et al. 2010 - https://pure.ulster.ac.uk/ws/files/11541925/HMW_Biosurfactants.pdf

Rudden et al 2015 - https://link.springer.com/article/10.1007/s00253-015-6837-1

Twigg et al, 2021 - https://sfamjournals.onlinelibrary.wiley.com/doi/full/10.1111/1751-7915.13704

Zompra et al. 2022 - https://pure.ulster.ac.uk/en/publications/multi-method-biophysical-analysis-in-discovery-identification-and

Adu et al. 2022 - https://www.mdpi.com/1999-4923/14/2/360

Callahan et al. 2022 - https://link.springer.com/article/10.1007/s00253-022-12115-6

  • These 2 additions would improve significantly the impact of the article.

  1. Also, why did the authors used a relatively new drug like piceatannol and not a more common and well-characterized drug, such as cisplatin or doxorubicin, as a control drug for the experiments?

As we state in the manuscript (ln 120), Piceatannol was not used as a control drug but as a comparison to an alternative naturally derived compound that has also been advanced as a potential anti-cancer agent. We have amended lines 119-120 and 475-477 to better reflect this point.

  1. Finally, minor editing errors are present and must be corrected (e.g., missing dot at line 298).

We thank the reviewer for pointing out this error. We have re-read the manuscript and have corrected minor errors such as this.

  1. The bibliography must be formatted properly according to the journal style (e.g. lines 84, 102, 103, 175, 184, etc.).

The bibliography in text citation have been modified as suggested.

  1. Rpm must be converted to either rcf or g (lines 124 and 171).

The “220 rpm” written in lines 129 and 178 is in reference to the platform orbital shaker speed for the incubator used for bacterial culture it was not for a centrifuge.

  1. Line 192: please insert the reference to the original article.

The original article the reviewer is refereeing regarding the methodology for the determination of surface tension and critical micelle concentration was originally published by Pierre Du Noüy in 1925. This reference was added as requested.

Finally, an overall editing to improve language presentation and references included.

Reviewer 3 Report

Microbial biosurfactants are amphiphilic secondary metabolites synthesised by bacterial and fungal species that possess surface-active properties. In this study, Twigg et al. reported that mono-rhamnolipids significantly reduced the viability of both colorectal cancer cell lines whilst having little effect on the healthy colonic epithelial cell line. Mono-rhamnolipids induced necrosis in both colorectal cancer cell lines. 

Major comments

The authors should perform animal experiments using Xenograft model to show the anti-cancer effects of Mono-rhamnolipids against colorectal cancer cells.

The authors should clarify the mechanism of action of anti-cancer effects of Mono-rhamnolipids.

Minor comment

In Fig.3, the authors should make the font size bigger. Moreover, the pictures of top right two look like the same.

Author Response

Microbial biosurfactants are amphiphilic secondary metabolites synthesized by bacterial and fungal species that possess surface-active properties. In this study, Twigg et al. reported that mono-rhamnolipids significantly reduced the viability of both colorectal cancer cell lines whilst having little effect on the healthy colonic epithelial cell line. Mono-rhamnolipids induced necrosis in both colorectal cancer cell lines. 

Major comments

The authors should perform animal experiments using Xenograft model to show the anti-cancer effects of Mono-rhamnolipids against colorectal cancer cells.

We thank the reviewer for their constructive criticism of our manuscript.

This is a proof-of-concept study and as such only in vitro cell models were used. Under the regulatory environment governing animal experimentation in Northern Ireland, it is highly unlikely that we would secure ethical approval for a xenograft model at this stage. As we have stated in the revised conclusion section of the manuscript, we do have plans to progress this work into an animal model however this again would be mainly reliant on regulatory and ethical approval.

The authors should clarify the mechanism of action of anti-cancer effects of Mono-rhamnolipids.

We state at the end of the manuscript, the study we present here was a preliminary proof-of-principle investigation of the potential anti-cancer activity of mono-RL congeners. The study is complementary to several publications we have generated over the past 4 years investigating the potential role of glycolipid-type biosurfactants in cancer treatment. As such the aim of the study was to demonstrate that these congeners negativity effect a cancer cell lines that would be accessible for a biosurfactant delivered either topically, orally, or anally. In this regard we feel the results we present here have fulfilled the study’s aim. We also present cellular staining data postulating a potential mechanism of cell death that is induced by these congeners. We accept that this is limiting, and we have now secured funding to further investigate the mechanisms of action using the various techniques proteomic/ metabolomic/ lipidomic techniques and hope to report these findings in the future. We have made alterations in the conclusion section of the manuscript to provide a more detailed description of our plans to progress this work up the technology readiness scale, please see lines 533-543.

Minor comment

In Fig.3, the authors should make the font size bigger. Moreover, the pictures of top right two look like the same.

The duplication of two panels in figure 3 was unfortunately an error. We apologize for and have corrected this error and for full disclosure sent the journal editor all raw data relating to the generation of this figure. We have also increased the font size in figure 3.

Finally, an overall editing to improve language presentation and references included.

Round 2

Reviewer 1 Report

The manuscript has been greatly improved. I am satisfied with the authors' responses to the comments. However, in response to the third comment, you mentioned "evidence" twice, but there was no corresponding content to support it. Please provide detailed information consistent with the content of your response to these two "evidence".

Author Response

We believe this point is in relation to the reviewer’s original comment “The ingredient is derived from bacteria, and the purity used in the experiment is not 100%. Is there sufficient data to prove the biological safety of the ingredient and the strain (Pseudomonas aeruginosa) and its secretions in the experiment?” To this end we stated that “Finally in addressing the reviewers concern of the use of the bacterium Pseudomonas aeruginosa to synthesize the compounds, this is something we accept may have to change to allow progression of the technology due to the pathogenic nature of the species. This could be achieved by genetically modifying a non-pathogenic host strain such as Pseudomonas putida to produce mono-RLs. There is evidence that this has been successfully carried out resulting a good yield of compound.”

We accept that we did not modify the manuscript to state this evidence. We have now carried out this modification citing literature that shows rhamnolipid by non-pathogenic pseudomonas species (Twigg et al., 2018 – ref 23), and mono-RL biosynthesis by recombinant E. coli and Pseudomonas putida strains (Cabrera-Valladares et al. 2006 – ref 54; Wittgens et al. 2011 – ref 55 respectively). Please see lines 500 -504 in the second resubmission of our manuscript.

Reviewer 3 Report

In this revised manuscript, the authors corrected the manuscript faithfully and appropriately according to the comments raised by the reviewers. By making these corrections, the manuscript is further improved and points that the authors would like to emphasize become much clearer.

Author Response

We thank the reviewer for their assessment of our revised manuscript.